# Neurodevelopment of 24 children born in Brazil with congenital Zika syndrome in 2015: a case series study

Lucas V Alves,[1] Camila E Paredes,[2] Germanna C Silva,[2] Júlia G Mello,[3] João G Alves[3]

[1]Department of Paediatrics Neurology, Instituto de Medicina Integral Professor Fernando Figueira, Recife, Pernambuco, Brazil
[2]Teaching Department, Faculdade Pernambucana de Saude, Recife, Pernambuco, Brazil
[3]Department of Paediatrics, Instituto de Medicina Integral Professor Fernando Figueira, Recife, Pernambuco, Brazil

**Correspondence to**
Dr João G Alves;
joaoguilherme@imip.org.br

## ABSTRACT

**Objective**  To describe the neurodevelopment of children with congenital Zika syndrome during the second year of life.

**Design**  Case series study.

**Setting**  Instituto de Medicina Integral Professor Fernando Figueira (IMIP), Pernambuco, Brazil.

**Participants**  24 children with congenital Zika syndrome born with microcephaly during the Zika outbreak in Brazil in 2015 and followed up at the IMIP during their second year of life.

**Main outcome measure**  Denver Developmental Screening Test II, head circumference and clinical neurological examination.

**Results**  All children presented neurodevelopmental delay: for an average chronological age of 19.9 months, language was equivalent to that of age 2.1 months, gross motor 2.7 months, fine motor/adaptive 3.1 months and personal/social 3.4 months. Head circumference remained below the third percentile for age and gender, and growth rate up to the second year of life was 10.3 cm (expected growth 13 cm). Muscle tone was increased in 23 (95.5%) of 24 children, musculotendinous reflexes were increased in the whole sample and clonus was present in 18 (77.3%) of 24 children. All children except one had epilepsy.

**Conclusion**  Children born with microcephaly associated with congenital Zika virus have a significant neurodevelopmental delay.

## INTRODUCTION

After the Zika outbreak in Brazil (2015/2016), healthcare professionals were faced with a population of children with congenital Zika syndrome, born with microcephaly and many neurological manifestations.[1 2] These children may present with epilepsy, abnormalities in tone or movement, including marked hypertonia and signs of extrapyramidal involvement, congenital limb contractures, dysphagia, sensorineural hearing loss, and visual involvement.[3 4] However there are no published follow-up studies reporting neurodevelopment, head growth and evolution of neurological manifestations in children with congenital Zika syndrome. Some of these children have been followed at the Instituto

### Strengths and limitations of this study

► A good-sized cohort of children with congenital Zika syndrome were followed up to 2 years of age.
► All children were born with microcephaly, and between the ages of 18 and 24 months underwent a complete neurological evaluation, including the Denver Developmental Screening Test II.
► Children with congenital Zika infection but were asymptomatic or with less severe symptoms at birth were not studied.

de Medicina Integral Professor Fernando Figueira (IMIP), and they are now around 2 years old. Our aim is to describe the neurodevelopment, head growth and neurological clinical manifestations of these children with congenital Zika syndrome during the first 2 years of life.

## METHODS

A case series study was conducted at the IMIP, Recife, Brazil, between January and August 2017. The IMIP is the largest hospital in northeastern Brazil. The first cases of Zika virus outbreak in Brazil associated with microcephaly were registered at the IMIP. From August 2015 to March 2016, 3440 children were born at the IMIP, and 178 had microcephaly associated with congenital Zika virus infection. At the time of this study 109 children born between this period with microcephaly and congenital Zika syndrome were being followed at the Department of Paediatrics Neurology of the IMIP. Twenty-four children were close to completing the second year of life and are presented in this study. These children were born with microcephaly and congenital Zika syndrome, as defined by the WHO (https://www.cdc.gov/pregnancy/zika/research/microcephaly-case-definitions.html5). Between the ages of 18 and 24 months, the children underwent a complete neurological evaluation, including the Denver Developmental Screening Test II.

**Table 1** Head circumference growth from birth to current age

| Gender | Current age (months) | Head circumference at birth (cm) | Current head circumference (cm) | Head circumference expected (cm) | Head growth achieved (%) |
|---|---|---|---|---|---|
| Female | 19 | 31 | 41 | 45.7 | 63.6 |
| Female | 18 | 30 | 38 | 45.6 | 51.2 |
| Male | 21 | 26 | 36 | 46.4 | 49.0 |
| Male | 20 | 27 | 39 | 46.3 | 62.5 |
| Female | 21 | 32 | 41 | 45.8 | 65.2 |
| Male | 22 | 23 | 37 | 47.5 | 57.1 |
| Female | 20 | 30 | 37 | 45.8 | 44.3 |
| Male | 19 | 31 | 44 | 47.2 | 80.2 |
| Male | 17 | 27 | 35 | 47.0 | 40.0 |
| Female | 20 | 29 | 43.5 | 45.8 | 86.3 |
| Female | 21 | 28 | 36 | 45.9 | 44.6 |
| Male | 21 | 29 | 41 | 46.4 | 68.9 |
| Female | 19 | 26 | 37.5 | 45.7 | 58.3 |
| Male | 19 | 29 | 45 | 47.2 | 87.9 |
| Female | 19 | 28 | 37 | 45.7 | 50.8 |
| Female | 19 | 29.5 | 39 | 45.7 | 58.6 |
| Male | 18 | 27 | 42 | 47.2 | 74.2 |
| Female | 19 | 31.5 | 42 | 45.7 | 66.9 |
| Female | 22 | 25 | 35.5 | 47.0 | 47.7 |
| Female | 22 | 31 | 40 | 47.0 | 56.2 |
| Male | 24 | 29 | 37.5 | 47.7 | 45.4 |
| Female | 19 | 30.5 | 42.5 | 45.7 | 78.9 |
| Male | 16 | 30 | 41 | 45.6 | 70.5 |
| Female | 24 | 31 | 43 | 47.0 | 75.0 |

All children were born with a head circumference below the third percentile for gestational age and gender. Congenital Zika syndrome was characterised by microcephaly associated with the following neuroimaging patterns: cerebral calcifications, ventriculomegaly, malformation of cortical development, hypoplasia of the cerebellum or brainstem, and abnormalities of white matter. Laboratory findings excluded STORCH (syphilis, toxoplasmosis, rubella, cytomegalovirus and herpes virus) infections in the mother or the baby, or both; and a serum or cerebrospinal fluid sample of the infant was positive for IgM anti-Zika.

A clinical evaluation included a complete physical and neurological examination (history, posture, cranial nerves, muscle tone, sensory and primitive reflexes), measurement of head circumference, and assessment of psychomotor development by checking if the main neurological milestones were appropriate for age. The Denver Developmental Screening Test II was also applied. This test is a screening tool used routinely in paediatric care to assess developmental milestones in children aged 0–6 years old. It has high inter-rater reliability and includes 125 items divided into four sections: psychosocial (aspects regarding the socialisation of the child inside and outside the family environment), language (production

of sound, and ability to recognise, understand and use the language), gross motor (body motor control, how to sit and walk) and fine motor/adaptive (eye–hand coordination, manipulation of small objects). The possible outcomes were classified as normal, delay and non-testable. Clinical exam and Denver Developmental Screening Test II were conducted by a neuropaediatrician specialist. To address potential sources of bias, the same neuropaediatrician (LVA) did multiple neurological examinations of the children followed throughout the 2 years.

### Patient and public involvement
All families were previously informed about the research question and the outcome measures. Patients were not involved in the development of plans for recruitment, design or implementation of the study. The results of the research were disseminated to study participants.

### RESULTS
Twenty-four children with congenital Zika syndrome were evaluated. Fourteen (58.3%) of the children were female, and their age ranged from 16 to 24 months (19.7 months). Mothers' age ranged from 15 to 39 years,

**Table 2** Denver Developmental Screening Test II in 24 children with congenital Zika syndrome

| Current age (months) | Language (equivalence in age months) | Gross motor (equivalence in age months) | Fine motor/adaptive (equivalence in age months) | Personal/social (equivalence in age months) |
|---|---|---|---|---|
| 19 | 1 | 1 | 1 | 1 |
| 18 | 1 | 4 | 4 | 4 |
| 21 | 1 | 4 | 4 | 3 |
| 20 | 1 | 1 | 1 | 3 |
| 21 | 4 | 4 | 4 | 4 |
| 22 | 1 | 1 | 1 | 3 |
| 20 | 1 | 1 | 1 | 1 |
| 19 | 6 | 7 | 7 | 6 |
| 17 | 1 | 1 | 1 | 1 |
| 20 | 7 | 7 | 7 | 6 |
| 21 | 1 | 1 | 1 | 3 |
| 21 | 1 | 4 | 8 | 8 |
| 19 | 1 | 1 | 4 | 4 |
| 19 | 1 | 1 | 1 | 3 |
| 19 | 4 | 5 | 4 | 4 |
| 19 | 4 | 1 | 1 | 4 |
| 18 | 1 | 1 | 1 | 1 |
| 19 | 1 | 1 | 1 | 1 |
| 22 | 4 | 4 | 6 | 6 |
| 22 | 3 | 4 | 4 | 4 |
| 24 | 1 | 4 | 4 | 3 |
| 19 | 1 | 3 | 3 | 3 |
| 16 | 2 | 2 | 3 | 3 |
| 24 | 2 | 3 | 3 | 4 |

and 10 of 24 (41.6%) were primiparous; 17 of 24 had a prenatal diagnosis of microcephaly by ultrasound. Some mothers reported fever (9/24; 40%) and rash (13/24; 54%) during the first half of pregnancy.

Two children were preterm (31 and 35 gestational weeks). All infants were fed by gastric tube, and 6 of 24 (25%) were malnourished.

Head circumference growth is shown in table 1. The mean growth of head perimeter from birth to the age of clinical evaluation was 10.3 cm. Eight (33.3%) of the 24 children had hospital admissions during their first 2 years of life: non-controlled epilepsy (2), ventriculo-peritoneal shunt (2) and infection (3), and diarrhoea, urinary tract infection and pneumonia. Almost all children (23/24) had recurrent episodes of convulsion and are receiving antiepileptic drugs for treatment.

Neurological evaluation showed that all children presented impairment of neuropsychomotor development; none stood with support, walked or were able to say a word. The results of the Denver Developmental Screening Test II according to the equivalent age are shown in table 2. Greater impairment was observed in the language section and lower impairment in the psychosocial section.

The mean age on clinical examination was 19.9 months, and the equivalent age for language, gross motor, fine motor/adaptive and personal/social was, respectively, 2.1 months, 2.7 months, 3.1 months and 3.4 months. All children attend a weekly session of motor physiotherapy, occupational therapy and speech therapy, and the families were also receiving psychological support.

Tone was increased in 23 (95.5%) of 24 children, musculotendinous reflexes were increased in the whole sample and clonus was present in 18 (77.3%) of the 24 children.

## DISCUSSION

A severe impairment of the neuropsychomotor development of children with congenital Zika syndrome was observed. The 24 studied children around 2 years of age could not stand alone, walk or say a word—these are expected developmental milestones for the studied age range. It should be noted that these children were followed up in a teaching hospital and were assisted by several health professionals, including physicians, physiotherapists, speech therapists, occupational therapists,

psychologists and nurses. This seems to assume that the prognosis of these children is very poor despite intensive therapeutic support.

Congenital microcephaly regardless of cause presents a significant risk for delay across all aspects of development and for long-term disability. Serious developmental delay was found in all 24 studied children with congenital Zika syndrome around 2 years of age. Gordon-Lipkin et al,[5] studying children with congenital microcephaly of different aetiologies, found that 16 (73%) of 22 had delay in development: gross motor (65%), fine motor (59%) and language (59%). Perhaps the fact that we had studied children with severe microcephaly explains this difference. In addition, the evolution of children with microcephaly associated with Zika virus is still unknown as this syndrome has only recently been described. Our report is a pioneer study describing the development during the first 2 years of life of children born with microcephaly associated with Zika virus.

All studied children had microcephaly and neuroimaging abnormalities. They also had positive serology for Zika virus and negative sorology for STORCH (Syphilis, Toxoplasmosis, Rubella, Cytomegalovirus and Herpes simplex) were excluded. Moreover, they did not present syndromic facies or congenital malformations. All these seem to confirm the Zika virus as an aetiological agent of our case series study. Besides microcephaly the head growth during the first months of life remained below the normal values (observed 10.3 cm and expected 13 or 14 cm), and all children kept a head circumference below the third percentile for age and gender.

Zika virus infects the neural progenitor cells, leading to less cell migration, neurogenesis impairment, cell death, and consequently microcephaly in newborns.[6] Additionally there is some evidence that Zika virus can continue to replicate in fetal brains during the first months of extra-uterine life.[7] All these seem to cause damage for the rest of their life.

Almost all studied children have epilepsy. According to a recent review, 54% of children with congenital Zika infection develop epileptic seizures during the first year of life.[8] Maybe this difference can be attributed to the fact that we only studied children with severe microcephaly and have followed these children over 2 years of age. Some studies have shown that convulsive epilepsy worsens the developmental delay of children with microcephaly.[9]

The Denver Developmental Screening Test II is culturally adapted to Brazil and has high inter-rater reliability.[10] It was applied to children of ages between 17 and 24 months, and the results obtained were equivalent to the development of those aged 1–8 months. The Denver II categorises a child's performance as 'Delay' (a child failing an element which ≥90% of children who are of his/her age would pass).[11]

The results of the Denver test in the present study compared with other studies with other types of congenital infections (Cytomegalovirus (CMV) and rubella) have shown more disappointing results.[12 13] This means

that a serious impact on development is associated with congenital Zika syndrome.

Our study had strengths and limitations. For the first time the neurodevelopment of infants born with microcephaly associated with Zika virus is reported. Unfortunately, due to operational difficulties, we were unable to study all 57 children followed at the IMIP and who are close to completing 2 years of age. It should be noted that many children used drugs to treat epilepsy and that this may interfere with responses to the Denver test. However the Denver Developmental Screening Test II was conducted only by one evaluator, and the delays in the marks of neurodevelopment were very clear. Additionally, the same neuropaediatrician did multiple neurological examinations throughout the 2 years the children were followed. Another limitation is that we only studied children with microcephaly, constituting severe congenital Zika syndrome. It is speculated that microcephaly is likely an endpoint of this devastating congenital infection. Long-term studies are needed to assess the clinical relevance of brain anomalies that are encountered and the neurodevelopmental sequelae in children with congenital Zika infection without microcephaly.

## CONCLUSION

In the sample evaluated children born with microcephaly associated with congenital Zika virus had a significant neurodevelopment delay during their second year of life. As this study was the first to describe the neurodevelopment of children with congenital Zika syndrome, other follow-up studies are needed to confirm these findings. We urgently need to optimise the provision of healthcare and improve the quality of life of these patients.

**Contributors** LVA and JGM contributed to conception of the work. LVA, CEP and GCS contributed to the acquisition of data. LVA, JGM and JGA contributed to the analysis and interpretation of data for the work. LVA is the guarantor. LVA, CEP, GCS, JGM and JGA drafted the work and revised it critically. LVA, CEP, GCS, JGM and JGA approved the version to be published.

**Funding** This research was supported by Bill & Melinda Gates Foundation and CNPq grant 439986/2016-8.

**Disclaimer** The lead author (LVA) affirms that the manuscript is an honest, accurate and transparent account of the study being reported; that no important aspects of the study have been omitted; and that any discrepancies from the study as planned have been explained.

**Competing interests** None declared.

**Patient consent** Parental/guardian consent obtained.

**Ethics approval** This study was approved by the IMIP Research Ethical Committee (CAAE 6167876.1.0000.5201).

**Provenance and peer review** Not commissioned; externally peer reviewed.

**Data sharing statement** Clinical data are available from the corresponding author at joaoguilherme@imip.org.br.

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
