## [Reviewer comments · BMJ Open]

ARTICLE DETAILS

TITLE (PROVISIONAL)	NEURODEVELOPMENT OF 24 CHILDREN WITH CONGENITAL ZIKA SYNDROME – CASE SERIES STUDY
AUTHORS	Alves, Lucas Paredes, Camila Silva, Germanna Mello, Julia Alves, João

VERSION 1 - REVIEW

REVIEWER	Marianne Besnard
REVIEW RETURNED	16-Jan-2018

GENERAL COMMENTS	This article describes the neurological outcome of patients with microcephaly linked with Zika virus perpartum infection. It just confirms that the neurodevelopment is very delayed, which is the issue for all severe congenital microcephalic patients, so its sole interest is to confirm it in patients with CZS. It should be reviewed for the English expression and many errors. The titer should specify the place and the period of the study. In the abstract, p2 l32 : accompanied. It should specify patients with microcephaly and CZS. In the methods, it should be interesting to specify if the mother was infected in the first, second or third trimester or was asymptomatic. Authors can precise how many children in their population are infected by zika : only the 24 ones or more, with other clinical signs? The criteria of the study population need to be clearer. It would be interesting to know if these patients can manage for feeding or are artificially fed and how is their growth. P5 l5 : iteMs, psYcHo-social It would be interesting to compare the Denver test with a population of microcephalic patients without CZS (CMV, toxoplasmose, rubella, alcohol,...). Also, authors should specify if the patients are under anti-epileptic treatment which might impact the scores. In the results : p5 l56 , psYcHo-social. P6 L1 : equivalent. P6L43 : different. P7L7 : presentS. P7L9 : lead. P7L19 : haD. P7L54 : other, needed.
--

REVIEWER	Luciano Pamplona de Góes Cavalcanti
REVIEW RETURNED	17-Jan-2018

GENERAL COMMENTS	The article is very interesting and well written. However, I recommend a revision of the language by a native. Pag 7. lines 3 to 17 - this excerpt seems unnecessary for discussion of the text. Pag 7. lines 49 to 54 - this conclusion is only for the sample evaluated. It should be made clear that the IMIP closely follows children with severe limitations and that probably those percentages of neurological limitations found can not be extrapolated to all children with microcephaly by zika virus. Table 1 - Insert a new column showing the percentage difference between the current and expected circumference. table 2 - include a caption stating what the numbers 1, 2, 3, 4...
--

REVIEWER	James D Brien PhD
REVIEW RETURNED	06-Mar-2018

GENERAL COMMENTS	Summary: Alves et al. evaluate 24 ~2 year old children, whom at birth were diagnosed with microcephaly and Congenital Zika syndrome by administering Denver test II and measuring head circumference. The authors found that all children experienced developmental delay in the areas of language, gross motor, fine motor, and personal/social skills. In addition, the children all ranked below the 3rd percentile in head circumference and had other accompanying medical problems including epilepsy and muscle tone defects. In summary, the authors show that children born with microcephaly and Congenital Zika syndrome experience significant delay in neurological development and other substantial health disparities. This study is extremely important and contains all the data necessary for publication. What needs to be improved is the description of the approach and how their data fits into the current literature. Broad comments:  1. The authors have a good-sized cohort of children who were diagnosed with microcephaly and Congenital Zika syndrome and were able to follow up after 2 years of age to assess an important question: are this children experiencing developmental delay and other health problems 2 years after initial diagnosis? While this initial study is important, it is important to evaluate children who present with less severe symptoms at birth, as Congenital Zika syndrome is defined as a wide range of abnormalities. The long term impact of this diagnosis will be an important question to explore. 2. The paper is written very tersely. There is minimal detail given and almost no expansion on the potential implications of their findings. The flow of the paper is rather choppy, with almost no transition from one thought to the next and several grammatical and spelling errors. There are several "paragraphs" that are composed of a literally a single sentence. Several statements are made without citation, where citation is necessary. I would suggest that the text go through substantial revisions. 3. The authors need to provide a brief piece of background on the Denver test. What is it, what does it measure, how are they using it. The authors also state they complete a clinical neurological test, although this is not described nor interpreted.
---

	4. Specific comments:  1. In the “strengths and limitations” section, the authors did not describe strengths or limitations; just simply what they did. 2. Page 4, line 46, Clarify who had CNS positive IgM results? Mother or child 3. Page 5 line 5 multiple spelling errors. 4. Page 5, line 43, the authors should clarify if they referring to hospitalizations during the course of the entire 2 years of the infants’ lives. How does this frequency of hospitalization compare to other infants in the area? 5. Page 6, line 26, the authors should clarify how many of the infants are receiving additional therapy, how long, etc. 6. Page 6, line 41, the paper the authors cite describe infants with microcephaly that have between 50-75% chance of being developmentally delayed by age 2, though the findings of this paper are 100%. The authors need to address the difference. 7. The authors need to develop a full discussion of the results, the consequences of their findings and how this fits into the current literature. In the description of the physical condition of the children the condition of a healthy child of that age should be provided for comparison. Describe the setting, locations, and relevant dates, including periods of recruitment, exposure, follow-up, and data collection Needs to be improved. The authors did not mention if mothers were diagnosed with ZIKV in a specific trimester. No mention of how subjects were recruited. There was mention of substantial therapy that these children are receiving that is not well described and somewhat glossed over. Clearly define all outcomes, exposures, predictors, potential confounders, and effect modifiers. Give diagnostic criteria, if applicable OK. More detail could be given on different outcomes of Denver II test. What constitutes “developmental delay” was not explicitly stated. Describe any efforts to address potential sources of bias OK. While they admitted that a single physician specialist administered all tests, they did not describe an effort to address this potential bias; e.g. multiple evaluators
--	---

VERSION 1 – AUTHOR RESPONSE

Reviewer: 1

Reviewer Name: Marianne Besnard

Institution and Country: Centre Hospitalier de Polynésie française

Please state any competing interests or state ‘None declared’: No competing interests declared

Please leave your comments for the authors below

The manuscript should be re-written by an English interpreter. It could be more detailed especially for the criteria of selected population. See more on the attached files.

This article describes the neurological outcome of patients with microcephaly linked with Zika virus perpartum infection. It just confirms that the neurodevelopment is very delayed, which is the issue for all severe congenital microcephalic patients, so its sole interest is to confirm it in patients with CZS. It should be reviewed for the English expression and many errors. The titer should specify the place and the period of the study. In the abstract, p2 l32 : accompanied. It should specify patients with microcephaly and CZS. In the methods, it should be interesting to specify if the mother was infected in the first, second or third trimester or was asymptomatic. Authors can precise how many children in their population are infected by zika : only the 24 ones or more, with other clinical signs? The criteria of the study population need to be clearer. It would be interesting to know if these patients can manage for feeding or are artificially fed and how is their growth. P5 l5 : iteMs, psYcHo-social It would be interesting to compare the Denver test with a population of microcephalic patients without CZS (CMV, toxoplasmosis, rubella, alcohol,...). Also, authors should specify if the patients are under anti-epileptic treatment which might impact the scores. In the results : p5 l56 , psYcHo-social. P6 L1 : equivalent. P6L43 : different. P7L7 : presentS. P7L9 : lead. P7L19 : haD. P7L54 : other, needed. Answer: The manuscript was re-written by an English-speaking colleague. More detailed of selected population were added. The place and the period of the study were added to the title. The word "accompanied" was corrected in the abstract and it was specified patients with microcephaly and CZS. It is not know if Zika virus infection happens during the first, second or third trimester of gestation; we only know that mothers reported fever (40%) and rash (70%) during the first half of pregnancy. The study population was detailed. All children were fed by gastric tube and 6/24 (25%) were malnourished. Thank you for this correction (psYcHo-social). A comment about Denver test in others congenital infections was added in the Discussion section. It was specified that all patients with epilepsy were under treatment and that this may interfere with the results of the Denver test. Many thanks for all corrections pointed out in the text.

Reviewer: 2

Reviewer Name: Luciano Pamplona de Góes Cavalcanti

Institution and Country: Universidade Federal do Ceará / Brasil

Please state any competing interests or state 'None declared': None declared

Please leave your comments for the authors below

The article is very interesting and well written. However, I recommend a revision of the language by a native.

Answer: The manuscript was reviewed by an English-speaking colleague.

Pag 7. lines 3 to 17 - this excerpt seems unnecessary for discussion of the text.

Answer: These paragraphs were changed.

Pag 7. lines 49 to 54 - this conclusion is only for the sample evaluated. It should be made clear that the IMIP closely follows children with severe limitations and that probably those percentages of neurological limitations found can not be extrapolated to all children with microcephaly by zika virus.

Answer: The conclusion was changed.

Table 1 - Insert a new column showing the percentage difference between the current and expected circumference.

Answer: This new column was added.

table 2 - include a caption stating what the numbers 1, 2, 3, 4...

Answer: This was done. This number represents the age in months corresponding to the Denver Developmental Screening Test II result.

Reviewer: 3

Reviewer Name: James D Brien PhD

Institution and Country: Saint Louis University, School of Medicine, USA

Please state any competing interests or state 'None declared': None declared

Please leave your comments for the authors below

Summary:

Alves et al. evaluate 24 ~2 year old children, whom at birth were diagnosed with microcephaly and Congenital Zika syndrome by administering Denver test II and measuring head circumference. The authors found that all children experienced developmental delay in the areas of language, gross motor, fine motor, and personal/social skills. In addition, the children all ranked below the 3rd percentile in head circumference and had other accompanying medical problems including epilepsy and muscle tone defects. In summary, the authors show that children born with microcephaly and Congenital Zika syndrome experience significant delay in neurological development and other substantial health disparities. This study is extremely important and contains all the data necessary for publication. What needs to be improved is the description of the approach and how their data fits into the current literature.

Broad comments:

1. The authors have a good-sized cohort of children who were diagnosed with microcephaly and Congenital Zika syndrome and were able to follow up after 2 years of age to assess an important question: are these children experiencing developmental delay and other health problems 2 years after initial diagnosis? While this initial study is important, it is important to evaluate children who present with less severe symptoms at birth, as Congenital Zika syndrome is defined as a wide range of abnormalities. The long term impact of this diagnosis will be an important question to explore.
Answer: It is a very important question but all children followed at IMIP had microcephaly and other signals of severe congenital Zika syndrome. We are trying to answer this important question in another study with children who were born during the Zika outbreak without microcephaly.

2. The paper is written very tersely. There is minimal detail given and almost no expansion on the potential implications of their findings. The flow of the paper is rather choppy, with almost no transition from one thought to the next and several grammatical and spelling errors. There are several "paragraphs" that are composed of a literally a single sentence. Several statements are made without citation, where citation is necessary. I would suggest that the text go through substantial revisions.
Answer: Thank you for this important comment. All the text was rewritten and reviewed by an English native colleague. We hope that we have complied with all your recommendations.

3. The authors need to provide a brief piece of background on the Denver test. What is it, what does it measure, how are they using it. The authors also state they complete a clinical neurological test, although this is not described nor interpreted.

Answer: A brief piece of background on the Denver development test II was provided; last paragraph of the Methods section. A complete clinical neurological examination was also described; last paragraph of the Methods section.

4. Specific comments:

1. In the "strengths and limitations" section, the authors did not describe strengths or limitations; just simply what they did.

Answer: Strengths and limitations section were re-written.

2. Page 4, line 46, Clarify who had CNS positive IgM results? Mother or child

Answer: It was clarified. IgM results were from the children.

3. Page 5 line 5 multiple spelling errors.

Answer: The spelling errors were corrected.

4. Page 5, line 43, the authors should clarify if they referring to hospitalizations during the course of the entire 2 years of the infants' lives. How does this frequency of hospitalization compare to other infants in the area?

Answer: It was added that hospitalizations were during the first 2 years of life. A comment of frequency of hospitalizations was added at the Discussion section.

5. Page 6, line 26, the authors should clarify how many of the infants are receiving additional therapy, how long, etc.

Answer: The additional therapy performed by children was added.

6. Page 6, line 41, the paper the authors cite describe infants with microcephaly that have between 50-75% chance of being developmentally delayed by age 2, though the findings of this paper are 100%. The authors need to address the difference.

Answer: One justification has been added.

7. The authors need to develop a full discussion of the results, the consequences of their findings and how this fits into the current literature. In the description of the physical condition of the children the condition of a healthy child of that age should be provided for comparison.

Answer: All these topics were now included in the Discussion section. New references were added and some were excluded.

Describe the setting, locations, and relevant dates, including periods of recruitment, exposure, follow-up, and data collection Needs to be improved. The authors did not mention if mothers were diagnosed with ZIKV in a specific trimester. No mention of how subjects were recruited. There was mention of substantial therapy that these children are receiving that is not well described and somewhat glossed over.

Answer: Information about setting and dates including periods of recruitment and follow-up were provided with more details. Unfortunately we do not know in which period of gestation the Zika virus infection occurred; at that time we had not yet the serological tests for Zika virus. It was added that all children attend a weekly session of motor physiotherapy, occupational therapy and speech therapy, as well all families were receiving psychological support. Also the information that children with epilepsy were receiving treatment with anti-epileptic drugs was provided.

Clearly define all outcomes, exposures, predictors, potential confounders, and effect modifiers. Give diagnostic criteria, if applicable OK. More detail could be given on different outcomes of Denver II test. What constitutes "developmental delay" was not explicitly stated.

Answer: Exposure was based on the intra-uterine infection by the Zika virus (maternal and fetal serology + STORCH exclusion) and a clinical picture compatible with congenital Zika syndrome (microcephaly and neuroimaging findings); diagnostic criteria was provided. Main outcome was neuromotor development based on Denver development screening test II. Head growth and neurological alterations (epilepsy, tonus, abnormal reflexes were secondary outcomes); all this information was also detailed. Developmental delay is categorized by the Denver screening development test II as "a child failing an element which $\geq 90\%$ of children who are his/her age would pass" (Frankenburg WK, Dodds J, Archer P, Shapiro H, Bresnick B. The Denver II: a major revision and restandardization of the Denver Developmental Screening Test. *Pediatrics*. 1992;89:91-7); this was added in the Discussion section including the reference).

Describe any efforts to address potential sources of bias OK. While they admitted that a single physician specialist administered all tests, they did not describe an effort to address this potential bias; e.g. multiple evaluators

Answer: To address potential sources of bias the same neuropsychiatrician (LVA) did multiple evaluations throughout the two years of follow-up.

VERSION 2 – REVIEW

REVIEWER	Marianne Besnard
REVIEW RETURNED	27-Apr-2018

GENERAL COMMENTS	the text is correct. We understand the severity of the microcephalic cases, whose neurodevelopmental delay is not surprising; it would be more interesting to evaluate later the outcome of the asymptomatic patients whose mother had a zika infection during pregnancy.
---

REVIEWER	James Brien
REVIEW RETURNED	01-May-2018

GENERAL COMMENTS	The authors have completed a significant level of additional work on the manuscript, generating a paper that is significantly easier and clearer to read.
---

VERSION 2 – AUTHOR RESPONSE

Reviewer(s)' Comments to Author:

Reviewer: 1

Reviewer Name: Marianne Besnard

Institution and Country: Centre Hospitalier de Polynésie Française

Please state any competing interests or state 'None declared': None declared

Please leave your comments for the authors below

the text is correct. We understand the severity of the microcephalic cases, whose neurodevelopmental delay is not surprising; it would be more interesting to evaluate later the outcome of the asymptomatic patients whose mother had a zika infection during pregnancy.

Answer: Thank you very much for your contribution to our study. The outcome of asymptomatic children whose mother had a Zika infection during pregnancy is being carried out and we hope to complete it this year.

Reviewer: 3

Reviewer Name: James Brien

Institution and Country: Saint Louis University, Saint Louis, Missouri USA

Please state any competing interests or state 'None declared': None declared

Please leave your comments for the authors below

The authors have completed a significant level of additional work on the manuscript, generating a paper that is significantly easier and clearer to read.

Answer: Thank you for your comments and suggestions that allowed us to greatly improve the quality of our manuscript.